# Complex Elucidation of Cells-of-Origin in Pediatric Soft Tissue Sarcoma: From Concepts to Real Life, Hide-and-Seek through Epigenetic and Transcriptional Reprogramming

**DOI:** 10.3390/ijms23116310

**Published:** 2022-06-04

**Authors:** Clara Savary, Cécile Picard, Nadège Corradini, Marie Castets

**Affiliations:** 1Childhood Cancer & Cell Death (C3), LabEx DEVweCAN, Centre Léon Bérard, Centre de Recherche en Cancérologie de Lyon (CRCL), Université Claude Bernard Lyon 1, INSERM 1052, CNRS 5286, 69008 Lyon, France; 2Department of Pathology, Hôpital Femme-Mère-Enfant, Hospices Civils de Lyon, Claude Bernard Lyon 1 University, 69002 Lyon, France; cecile.picard@chu-lyon.fr; 3Department of Pediatric Oncology, Institut d’Hematologie et d’Oncologie Pédiatrique, Centre Léon Bérard, 69008 Lyon, France; nadege.corradini@ihope.fr; 4Department of Translational Research in Pediatric Oncology, Centre Léon Bérard, 69008 Lyon, France

**Keywords:** sarcoma, soft tissue sarcoma, transcriptional networks, cellular reprogramming, transdifferentiation, cell-of-origin, epigenetics, tumor heterogeneity, clinical management

## Abstract

Soft tissue sarcoma (STS) comprise a large group of mesenchymal malignant tumors with heterogeneous cellular morphology, proliferative index, genetic lesions and, more importantly, clinical features. Full elucidation of this wide diversity remains a central question to improve their therapeutic management and the identity of cell(s)-of-origin from which these tumors arise is part of this enigma. Cellular reprogramming allows transitions of a mature cell between phenotypes, or identities, and represents one key driver of tumoral heterogeneity. Here, we discuss how cellular reprogramming mediated by driver genes in STS can profoundly reshape the molecular and morphological features of a transformed cell and lead to erroneous interpretation of its cell-of-origin. This review questions the fact that the epigenetic context in which a genetic alteration arises has to be taken into account as a key determinant of STS tumor initiation and progression. Retracing the cancer-initiating cell and its clonal evolution, notably via epigenetic approach, appears as a key lever for understanding the origin of these tumors and improving their clinical management.

## 1. Introduction

Soft tissue sarcoma (STS) represent a heterogeneous group of malignant tumors comprising a collection of more than 100 histological subtypes [1]. STS accounts for 1% of all adult solid malignant tumors and between 5% and 10% of all pediatric cancers [2]. These tumors can arise in a wide range of anatomical sites deriving from the mesenchymal lineage including adipose and connective tissues, muscles, tendons, nerves, vessels, synovial and stromal supporting cells [3]. Although mounting evidence supports that sarcomagenesis is the result of genetic alterations in mesenchymal progenitor/stem cells (MSCs), this notion remains unclear given the diversity of sources of these MSCs and the precise cellular origin of STS remains largely unknown [4].

STS display a wide range of clinical behaviors with varying metastatic potentials, which represent the most powerful predictor of outcome in patients [5]. Tumors with minimal metastatic potential (low-grade) are more likely to be cured with complete surgical resection, whereas tumors with a tendency for widespread metastatic dissemination (high-grade) have a higher risk of recurrence and dissemination following local therapy. Surgery then constitutes the primary treatment in STS, but is often combined with chemotherapy and/or radiotherapy, in particular for patients with unresectable or metastatic presentation [6,7]. This multimodal strategy increased the long-term survival of patients with localized STS (overall 5-year survival of about 80%), while the outcome for those with spread diseases remains dismal (overall 5-year survival of about 15%) [8]. This poor prognosis is explained by metastatic STS being refractory to radiation and chemotherapy treatments, which is of great concern as it represents one-third of overall patients [1]. The complex level of heterogeneity of STS constitutes an obstacle to improve their therapeutic management. Indeed, the response to conventional treatments varies greatly and cannot be translated either between different STS subtypes or between patients of the same subtype [9]. It is therefore necessary to better understand the molecular and cellular mechanisms underpinning the heterogeneity of STS in order to improve their clinical prognosis.

The inter-patients heterogeneity observed in STS could result from the acquisition of different oncogenic modules but could also reflect their different cells-of-origin [10,11,12]. The cell-of-origin is defined as the normal cell that acquires the first cancer-promoting mutation(s) and refers to a cancer-initiating or tumor-initiating cell [13]. The cancer biology has been conceptualized by a complex interplay between non-mutually exclusive mechanisms involving molecular events (genetic) and the cell-of-origin (epigenetic). This is illustrated by (1) distinct oncogenic events occurring within the same target cell leading to different tumor phenotypes; and (2) the outcome of a given genetic alteration that can differ depending on the epigenetic context of the cell in which it arises [13,14,15]. Accumulating evidences show that the epigenetic state of a cell, defined by its chromatin landscape, appears to be crucial in providing a permissive milieu for context-specific tumorigenesis [4,16,17,18]. Therefore, investigating the primary cells that are permissive to oncogenic drivers could pave the way for identifying the spectrum of cell-of-origin of STS, and shed light on the impact of this genetic/epigenetic crosstalk in determining tumor fate.

The objective of this review is to show the complexity of identifying the cell of origin of STS, and how far we still have to go considering the importance of this factor in patients’ therapeutic management. We focus on how the mechanisms of cellular reprogramming, driven by a complex interplay between genetic and epigenetic changes can lead to misleading definition of the cell-of-origin of STS by relying solely on gene expression profiles or histological markers of normal differentiation. Using the example of rhabdomyosarcoma, the most common form of STS in children and adolescents, we describe how cellular origin can impact cancer evolution and contribute to inter-tumoral heterogeneity of STS. We argue that analysis of heritable epigenetic marks constitutes a complementary strategy to overcome the cellular reprogramming confusion. Finally, we illustrate the clinical relevance of accurate inference of cell(s)-of-origin to improve the clinical management of patients with STS.

## 2. Epigenetic Alterations and Context in STS: Impact on Oncogenic Reprogramming

Neoplastic transformation involves a profound cellular reprogramming in which fully differentiated and functional cells lose aspects of their identity, while gaining progenitor characteristics (dedifferentiation) or adopt distinct differentiated state (transdifferentiation) [16,19,20]. Initially, it was accepted that “once a cell has concluded its differentiation path towards a specific fate, this state is permanent and irreversible” [16], but this statement has been overturned by the cell plasticity concept, which designates the ability of mature cells to switch phenotype or identity [19,21]. This plastic process can result from random genetic and epigenetic remodeling, and is particularly relevant as it plays a pivotal role in tumor initiation, progression, therapeutic resistance and relapse [4,19,21,22,23,24]. Understanding the reprogramming mechanisms involved in the etiology of STS and their diversity requires the integration of three variables including: the underlying genetic alterations, the epigenetic context and the temporal evolutionary dynamics of both parameters.

### 2.1. Epigenetic Oncogenic Driver Events and Their Role in STS Etiology

The field of sarcoma biology is already familiar with powerful oncogenic events that can achieve a profound cellular reprogramming. Sarcomagenesis is driven by a wide spectrum of genetic alterations used to classify sarcomas into two main categories: (1) those with simple karyotypes’ defects, including recurrent chromosomal translocations, chromosomal amplifications, and specific activating or silencing oncogenic mutations; and (2) those with complex karyotypic defects, characteristic of severe genetic and chromosomal instability, with no tumor-specific genetic alterations [4,7,25,26,27,28].

Chromosomal translocations are a key determinant of oncogenic reprogramming in STS, since about one-third of all cases are found to harbor balanced chromosomal translocations [3,29]. These translocations often lead to fusion of two distinct transcription factors-encoding genes’ portions, the DNA binding domain of one, and the transactivation domain of the other [4,5,7,30,31,32,33,34,35,36,37,38,39,40]. Such chimeric fusion proteins present higher transcriptional activity than their wild-type counterparts and, in most cases, exhibit altered target genes specificity. Their abnormal pattern of expression can achieve drastic changes in transcriptomic and epigenetic landscapes and are believed to trigger reprogramming in appropriate permissive cells [4,7]. For example, the unbalanced translocation t(X;17)(p11.2;q25) is characteristic of alveolar soft part sarcoma and identified in the majority of these tumors [41,42]. The N-terminal portion of *TFE3* is being replaced by *ASPSCR1* (*ASPL*) sequences, while retaining its DNA-binding domain, and is implicated in transcriptional deregulation and pathogenesis of this type of sarcoma [41]. Among the most studied examples, we can also mention the fusion of the DNA-binding domain of *PAX3* or *PAX7* transcription factors with the transactivation domain of *FOXO1* (*FKHR*) in rhabdomyosarcoma (RMS) that is directly associated with gene expression dysregulation [35,43]. Similarly, unique transcriptomic signatures are observed in each type of small round cell sarcomas, depending on their driver fusion gene (i.e., CIC- or BCOR-fused [44]). However, although major in case of chromosomal translocation affecting transcription factors, oncogenic transcriptional reprogramming also exists in STS without fusion transcripts, and can be induced by single gene mutations. This notably holds true when mutation affects transcriptions factors, such as the L122R gain-of-function mutation in *MYOD1* observed in spindle cell/sclerosing rhabdomyosarcoma, or in the *CTNBB1* mutations that are recurrent in desmoid tumors for example [45,46]. These particular signatures can serve as a guide to improve the molecular classification of this complex entity.

Besides those transcriptional rewiring, direct perturbations in the epigenetic machinery have emerged as driver mechanisms of tumorigenesis in various STS and have been found sufficient to induce oncogenic reprogramming and transdifferentiation [28,47,48,49,50]. Epigenetics is critical in establishing and maintaining cell-type identity and its dysregulation can lower the barriers for transition between cell states [15,51,52]. The differentiation of a cell is driven by sequential genes expression patterns that largely depends on chromatin accessibility [13,53,54,55,56]. Indeed, physical interactions between chromatin-binding factors and regulatory genomic regions (enhancers, promoters, insulators) are necessary to cooperatively modulate genes expression. The organization of accessible chromatin across the genome reflects a permissive state for initiating transcription and is controlled by different epigenetic layers of regulation: DNA methylation, histones’ post-transcriptional modifications, nucleosome remodeling and subsequent modulation of the 3D chromatin structure [28,57]. This chromatin landscape is modulated by a collection of enzymatic proteins and complexes that can “write”, “read” or “erase” some epigenetic marks, such as acetylation and methylation, resulting in transcriptionally active, inactive or neutral states [58,59].

Dysregulation in post-transcriptional modification of histone represents a common mechanism for tumorigenesis in several subtypes of STS, such as malignant peripheral nerve sheath tumors (MPNSTs) [60,61,62,63], endometrial stromal sarcomas (ESSs) [64,65,66] or small blue round cell sarcomas [49,67,68,69]. The polycomb repressive complex (PRC) designs a group of chromatin-modifying proteins assembled into two canonical complexes, PRC1 and PRC2, which display respectively histone ubiquitin ligase (H2AK119ub) and methyltransferase (H3K27me3) activities [70,71]. In MPNSTs, two exclusive loss-of-function alterations have been reported in PRC2 core components: suppressor of zeste 12 (SUZ12) and embryonic ectoderm development (EED) [60,72]. As a result, EED and SUZ12-deficient MPNSTs exhibit chromatin remodeling modifications that lead to an upregulation of genes expression compared with their wild-type counterparts, directly involved in the transformation of normal cells. In addition, impaired function of PRC2 has also been described in ESSs as a consequence of the expression of the fusion genes *JAZF1–SUZ12* [72], *JAZF1-PHF1* [73] and *MBTD1–EZHIP* [74], which reduces the methyltransferase activity of the complex. In a subset of high-grade ESSs and small blue round cell tumors, internal tandem duplication and chromosomal translocations alter the activity of the BCL6 co-repressor (BCOR), a member of the PRC1.1 complex (a non-canonical PRC1) [65,66], which is involved in the maintenance of pluripotency in stem cell populations and in the suppression of mesodermal transcriptional programs [28]. Several studies support the fundamental role of PRC1 and PRC2 in preserving cell-type identity and the impact of their dysregulation on cellular reprogramming [75,76]. Therefore, genetic alterations in core components of PRC1 and PRC2 likely represent a profound epigenetic reprogramming event in STS, but their impact on dedifferentiation or transdifferentiation needs to be further elucidated.

Similarly, impaired functions of nucleosome-remodeling complexes, such as the SWI/SNF family, are implicated in the acquisition of a stem-like phenotype and in the oncogenic transformation in various subtypes of STS [77,78,79,80]. Nucleosome-remodeling complexes are specialized multi-protein machines that use the energy of ATP hydrolysis to alter the structure, composition, and position of nucleosomes, enabling access of transcriptional factors and initiators to the underlying DNA. The SWI/SNF family of chromatin remodelers (also known as BRG1/BRM associated factor (BAF) complexes) exists in at least three forms: canonical BAF (cBAF), polybromo-associated BAF complex (PBAF), and GLTSCR1/1L-containing BAF (GBAF; non-canonical BAF or ncBAF). Each of these large complexes are composed of approximately 15 protein subunits, including SMARCB1 (SNF5, BAF47 and INI1), SMARCC1/SMARCC2 (BAF155 and BAF170), and one of the two mutually exclusive ATPase subunits, SMARCA4 (BRG1) and SMARCA2 (BRM). Recent studies revealed that the SWI/SNF complex is commonly altered by genetic alterations in about 20% of all tumors [81]. Sarcomas make no exception with *SMARCB1* loss being characteristic of epithelioid sarcoma and malignant rhabdoid tumors [28,82]. In synovial sarcoma, the incorporation of SS18–SSX fusion protein into cBAF, instead of wild-type SS18, results in the ejection of SMARCB1 and its subsequent proteasome-mediated degradation [83].

Then, beside targeted activation of transcriptomic programs by chromosomal translocations involving transcription factors, epigenetic alterations appear to be a key determinant of the cellular reprogramming of some STS, which must be addressed given its importance in the deployment of specific oncogenic programs.

### 2.2. Permissivity of the Epigenetic Context for Cellular Reprogramming

Besides direct perturbation of its organization, there is mounting evidence that the epigenetic state of a cell is important in providing a permissive milieu to oncogenic transformation and may predetermined the outcome of genetic change(s) [11,53]. Indeed, activation of the same oncogenic pathway in different cellular compartments, or cellular contexts, can result in distinct tumor types [11,84]. For example, *EWSR1-ATF1* or *EWSR1-CREB1* chromosomal translocations can give rise to both clear cell sarcoma and angiomatoid fibrous histiocytoma, which are two STS subtypes with distinct morphological and clinical features (Figure 1A) [39,85,86]. Moreover, the t(X;17)(p11.2;q25) translocation resulting in *ASPL-TFE3* fusion gene can result in alveolar soft part sarcoma [41], but also a distinctive subset of renal cell carcinoma, which frequently has papillary architecture [87].

Along the same line, the use of transgenic mice models has improved our understanding of the importance of the epigenetic context, in which a given genetic alteration occurs. For example, expression of the pathognomonic *PAX3-FOXO1* fusion gene associated with p53 loss is sufficient to trigger RMS occurrence when expressed in Myf6+ fetal myogenic progenitors but not in Myf6+ postnatal committed myogenic progenitors (Figure 1B). Moreover, whereas this chromosomal translocation gives rise to tumors with an alveolar histology in these Myf6+ fetal precursors, its expression in Pax7+ post-natal satellite cells leads to the appearance of RMS tumors with pleomorphic characteristics [88]. Similarly, a model of liposarcoma-like tumors was successfully developed with transgenic mice expressing the *FUS–DDIT3* fusion gene alone [4,89,90], but identifying the appropriate cellular environment supporting its expression to generate myxoid liposarcoma still represents an active area of research. An additional level of complexity comes from the fact that the epigenetic landscape of a cell changes along with its differentiation stage [52,54,55]. Therefore, the permissiveness of the epigenetic context of one cellular lineage can change along with its developmental trajectory. This could partly explain why the onset of some tumors occurs sometimes in a very narrow period of time during childhood. As an example, three-quarters of the patients diagnosed with atypical teratoid/rhabdoid tumors are 3 years old or younger [91]. The issue is then not only to define which cells are susceptible and tolerant to a given oncogenic alteration, but also the permissive timepoints along their respective differentiation process.

Overall, the wide diversity of genetic events and epigenetic contexts of the cell(s)-of-origin give rise to a collection of molecular combinations, likely contributing to the large biological and clinical heterogeneity of STS. This shades light on how the lineage programs inherent to the tumor precursor cell is determinant in its susceptibility to oncogenic transformation and how it can influence the tumor cell fate and pathology. This reinforces the need to advance into the stratification of STS not only according to their genetic abnormalities but also to their cell-of-origin [13].

## 3. Cellular Reprogramming as a Source of Confusion in the Definition of the Cell-of-Origin in STS

### 3.1. Histologic and Transcriptomic Analogies: The Roots of STS Classification

The inference of cancer cell-of-origin has long been driven by histological and morphological resemblances with normal cell differentiation and has guided cancer classification for almost a century [18,53,92,93]. Indeed, tumor cells morphology and histology, as well as intra-tumoral hierarchy, reflect the differentiation trajectories that are deployed within tumors and are used to extrapolate the identity of the cell-of-origin. Consequently, STS are named eponymously after their presumed cell-of-origin, which historically relied on their histological appearance and anatomic tissue compartments [94]. However, this approach is complex for some entities, such as clear cell sarcoma and an angiomatoid histiocytofibroma, with non-specific or misleading histology, hence requiring additional techniques for diagnosis [3,85]. Immunohistochemistry first emerged as a powerful tool to identify diagnostic markers of poorly differentiated tumors, such as in RMS, and to propose hypotheses of originating cell [3,63,95,96,97]. More accurate inference of cell-of-origin have relied on comparisons between gene expression signatures of tumor and normal cell populations from which they may arise [53]. Recent studies have demonstrated that tumor subsets share transcriptomic similarity with their corresponding lineage of origin [53,98]. These findings support the view that developmental programs are recapitulated in many diverse solid tumor types, such as medulloblastoma [99] and melanoma [100]. Single-cell RNA sequencing (scRNA-seq) data have reinforced the interest in these comparative transcriptomic approaches as tools to characterize the lineage of origin of tumor cells by trajectory inferences analyses [101,102]. Moreover, even poorly differentiated tumor cells might preserve a lineage memory that reflects their developmental history [103,104]. Transcriptome-wide profiling helped clarify the complexity of STS classification by defining homogeneous molecular subgroups and gained insights into the molecular basis of poorly differentiated tumor types. For example, expression-profiling of synovial sarcoma indicate that these tumors are most closely related to MPNST tumors from a transcriptomic point of view, suggesting a possible neural crest origin [105,106]. To sum up, the gene expression signatures revealed by transcriptome-based techniques are believed to reflect “shadows” of the cell-of-origin in cancer.

### 3.2. Misleading Appearances and Confounding Factors in Defining the Cell-of-Origin of STS

However, initiating oncogenic events, and accumulating changes in the genetic and epigenetic cellular landscape, can lead to a profound cell identity crisis, sustained by reactivation of developmental or differentiation programs from completely distinct lineages, to support tumor progression [13,16,107,108,109]. As a result, the cancer cell transcriptome that shape the morphological features and phenotype of a cell then no longer resembles its cell-of-origin [11,99,110,111]. Then, some of the cell-of-origin attributions based on morphologic resemblance observations remain intact today, while others are named erroneously [3]. For example, synovial sarcoma was named for its resemblance to synovium, even if there is currently no argument to support that they derive from synoviocytes. RMS is a heterogeneous entity, with as a sole common denominator, embryonic myogenic attributes. Consequently, they have long been assumed to derive solely from muscle progenitors and precursors, but this view has been challenged recently. Indeed, although RMS exhibit markers of muscle differentiation and are most often found in tight proximity with skeletal muscle beds, there is growing evidence that the cell causing these cancers is not limited to skeletal muscle embryonic or stem cells. First, these tumors arise from many anatomic sites, including organs that are free of skeletal muscles, such as the bladder, prostate, salivary glands, and biliary tract [102,112]. Furthermore, animal models and xenografts of transformed human cells demonstrate that both skeletal muscle precursors [113,114] and non-myogenic lineage cells, such as endothelial cells or even MSCs, can give rise to RMS-like tumors [115,116]. Even more surprisingly, the expression of *PAX3-FOXO1* fusion gene in neural progenitors of chick embryos is sufficient to give rise to tumor masses, and to drive the establishment of a myogenic signature from a non-muscle lineage [117].

### 3.3. Cell-of-Origin of STS: Behind the Mesenchymal Origin Paradigm

STS are gathered as an entity based on their mesenchymal origin. However, the diversity of mesenchymal lineages and their pleiotropic developmental origins question the significance of this grouping in terms of molecular etiology. This can be illustrated with MSC, which have been proposed as candidate cell-of-origin for sarcomagenesis [118,119,120]. MSCs are multipotent cells with multiple characteristics including the ability to (1) give rise to a wide range of mesenchymal cell types including adipocytes, chondrocytes, skeletal myoblasts, osteocytes, neural cells, and fibroblasts, (2) adhere to plastic substrate in vitro. They also express a specific set of surface antigens markers. However, these inclusion criteria mask the heterogeneity of MSCs, which can originate from different mesodermal and neuroectodermal progenitors/precursors from different tissue sources at different time points during development/ontogenesis, each type being associated with a potentially particular epigenetic state [118,119,120,121,122]. Therefore, the question is less to define whether MSCs can be cells-of-origin of STS, but to identify which are the MSCs that are permissive to the expression of oncogenic drivers and characteristic of each of these types of cancers.

Then, it appears that relying on the gene expression profiles and histological/functional markers of a tumor cell can lead to erroneous inference of its cell-of-origin, or to insufficient characterization of its identity to extrapolate its behavior upon oncogenic transformation. Novel strategies are therefore needed to accurately identify the cell-of-origin of STS.

## 4. Epigenetics as a Powerful Tool to Refine the Cell-of-Origin of STS

Epigenetic marks are increasingly interpreted to infer cancer cell-of-origin as they may retain a ‘‘fossil record’’ of tissue-specific developmental programs, faithfully propagated from the first transformed cell throughout tumor progression [15,59,123,124,125]. During the developmental process, heritable epigenetic marks are required to define and maintain unique gene expression patterns, crucial for cell-type identity [124]. One of the best characterized epigenetic mark is DNA methylation, which usually refers to modified nucleotide 5-methylcytosine (5mC), and which has a key role in stabilizing the inheritance of genes expression responses across cell division. In mammals, the primary target for DNA methylation is the cytosine of CpG sites that are genomic sequences of dinucleotide cytosine(C)-phosphate(p)-guanine(G). Genomic regions exhibiting elevated frequencies of these CpG sites are known as “CpG islands” and are present at over two-thirds of gene promoters [126]. Recent findings show that hypomethylated CpG sites can preserve a nearly complete archive of their developmental origin [13,28,123]. For example, a recent study demonstrated that during retinogenesis, the most dramatic change was the derepression of cell-type-specific differentiation enhancers, while some progenitor and cell cycle genes remained epigenetically silenced [127]. Increasing evidence supports that this also holds true in cancers and that embryonic development’s epigenetic memory may be retained during cancer initiation and progression, despite genetic and epigenetic changes [123]. Accordingly, a recent study showed that retinoblastoma epigenome resembled a particular stage of the retina development [127]. Moreover, analysis of multiple samples regions of primary tumor, metastases and pre-malignant outgrowths in prostatic adenocarcinoma revealed that lineage tracing via DNA methylation heterogeneity closely mirrored the phylogenetic relationships built on copy number genetic diversity [15]. Similarly, a study of single-cell-derived colon cancer organoids demonstrated that marked DNA methylation heterogeneity was propagated stably, in parallel with genetic diversification [15]. Considering, the retention in tumor cells of epigenetic fingerprinting of the tissue of origin, DNA methylation is now used to infer the tissue of origin of cancers of unknown primary (CUP) sites [128]. In other words, genome-wide DNA methylation prints can reflect a tissue-specific developmental program and pave the way to better understand the cell-of-origin in many cancer types [129,130,131,132].

The question of STS cells-of-origin was mainly addressed using models, by inducing the expression of specific oncogenic drivers in a given precursor. The informational potential of epigenetic memory remains less explored [3]. DNA methylation profiles of STS was proven useful to improve lineage classification and to reveal methylome patterns that were specific of tumor types or stages [3,28]. These results allowed the distinction among subtypes of RMS [133], angiosarcoma [134] or small blue round cell sarcomas [135] for example. In addition, specific methylation profiles correlated with diverse clinical outcomes in dedifferentiated liposarcoma [96]. DNA methylation profile was also used to show that phenotypical differences between undifferentiated endometrial carcinoma and SMARCA4-deficient uterine sarcoma may result from SWI/SNF deficiency occurring in different cellular contexts. Overall, the subtype-specific DNA methylation patterns in STS suggests that they might arise from different timepoints of a same differentiation trajectory, or from possible distinct lineages, but proof-of-evidences are required. Based on this concept of “epigenetic memory”, further research using genome-wide DNA methylation analyses and their cross-referencing with transcriptomic data are therefore needed to better understand the cellular origin of STS (Figure 2).

## 5. Clinical Relevance of the Cell-of-Origin in the Management of STS

Beyond simply improving our knowledge of sarcoma biology, defining the identity of the lineage or cell(s) at the origin of the different histological and molecular types of STS is a key clinical issue. Indeed, if the epigenetic context clearly participates in defining the transformation capacity of a cell, it also constitutes an important determinant of treatment response and resistance [136,137,138]. Along this line, the work performed by Abraham and colleagues [88] has shown that the introduction of the *PAX3-FOXO1* translocation in Myf6+ fetal myogenic progenitors or in Pax7+ post-natal satellite cells results in the appearance of RMS-like tumors with singular level of sensitivity to cell cycle inhibitors, consistent with their respective degree of myogenic differentiation. Moreover, preliminary evidence already supports this notion of ‘context-driven’ therapeutic indices, in which combinations of genetic alterations and specific lineages constitute unique vulnerabilities. As an example, the presence of hemizygous deletions on chromosome 1p predict enhanced chemosensitivity in anaplastic oligodendrogliomas, whereas no correlation is found with tumors from other lineages [53,139]. Moreover, clinical trials show that a same drug targeting oncogenic BRAFV600E mutation is efficient in melanoma, non-small-cell lung carcinoma and hairy cell leukemia, but not in colorectal cancer [13,140,141,142]. More than targeting a given genetic abnormality, the next challenge in the therapeutic management of cancers, with a high degree of heterogeneity such as STS, will be to integrate the cell types and cell states from which they arise into clinical practice [88].

Indeed, theThe integration of the epigenetic component is essential to precisely define the Achilles’ heel of tumor cells and the actionable levers according to the cellular context to sensitize them to treatments. Thus, manipulation of the expression level of the *PAX3-FOXO1* transgene by using the histone deacetylase inhibitor entinostat potentiates the effect of actinomycin D, only when this fusion gene is expressed in Pax7+ satellite cells in the postnatal period [88]. If this statement needs to be tempered considering the important epigenetic reorganization undergone by tumor cells during the transformation and escape oncogenic process, which reduces the impact of the original cellular context, the approaches of personalized medicine should probably be rethought beyond the framework of genetics alone [133,143,144].

Another important aspect of elucidating STS cell(s)-of-origin concerns the identification of the environmental factors that may play a causal role in their initiation/escape. The developmental context in which childhood cancers occurs could *a priori* confer susceptibility to carcinogens exposure. Several environmental factors have been suggested to be associated with increased risk of several childhood cancers, but the formal demonstration of their implication as well as the mechanisms and molecular bases linking early life factors to childhood cancers remain poorly understood [145,146,147,148,149]. Environmental factors could exert an oncogenic action not only by mutagenesis but also by inducing epigenetic remodeling [150,151,152] and an accumulation of evidences indicates that exposures in utero, or in early life, induce significant epigenetic alterations. However, the impact of pollutants can differ depending on the cell type on which it acts. For example, it has been shown that neural crest cells (NCCs), from which several types of sarcoma are thought to derive, are very sensitive to environmental exposures. Prenatal exposure to tobacco, arsenic, or pesticides are all associated with defects in the formation of NCCs, and notably alter their differentiation, and their migration, to their proper location [153,154,155,156,157]. Beyond these migratory defects, these pollutants have been shown to alter the expression profile of genes that control NCCs fate, thereby altering the cellular context in which an oncogenic mutation can occur [157,158]. Consequently, the identification of the cell(s)-of-origin of STS becomes a key issue not only in therapeutic terms, but also from a prevention perspective, to better identify the impact of the exposome during embryonic development or ontogeny (Figure 3).

## 6. Conclusions and Future Perspectives

Cellular reprogramming, following oncogenic event(s), can profoundly reshape the transcriptome and epigenome of a transformed cell and can result in misinterpretation of the cell-of-origin of STS. Functional studies have shown that oncogenic transformation of cells from the muscular lineage, but also that non-muscle lineage (endothelial and neural progenitors), can lead to the emergence of cancer cells expressing striated muscle markers (i.e., RMS-like cells). This questions the stratification of STS, largely based on the differentiated cells they most resembled. Thus, accurate definition of the sarcoma cell-of-origin could help refine the current classification of STS.

Beside patients’ stratification, the precise cell type in which the first oncogenic event occurs is of crucial importance as it can influence the tumor phenotype and aggressiveness. A better understanding of the primary cell initiating sarcomagenesis could be translated into clinical use and improve the prognostic prediction of patients with STS.

Thus, the integration of genetic, transcriptomic and epigenetic data is likely to be necessary to improve our understanding of the complexity of these tumors’ biology and the medical management of sarcoma patients.

## Figures and Tables

**Figure 1 ijms-23-06310-f001:**
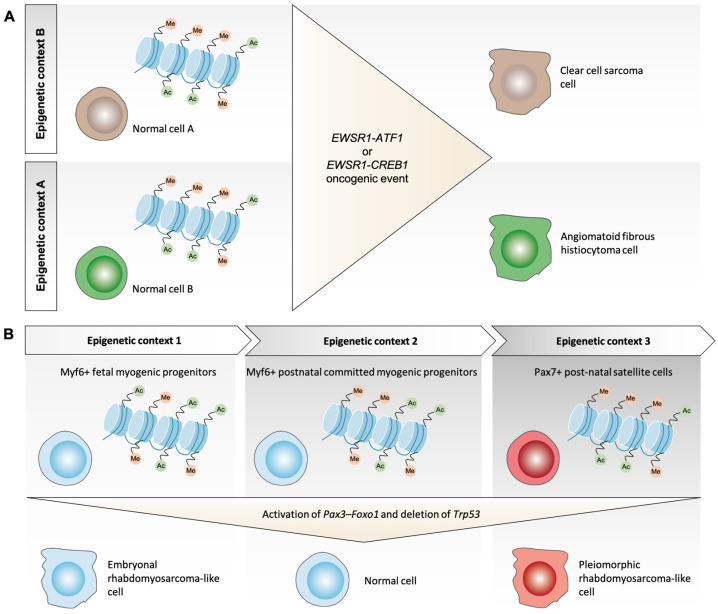
Permissivity of the epigenetic context for cellular reprogramming. (**A**) Expression of EWSR1-fusion genes leads to different phenotypical outcomes depending on the cell in which it occurs. (**B**) Pax3-Foxo1 oncogenic transformation power relies on the cell state in which it is expressed.

**Figure 2 ijms-23-06310-f002:**
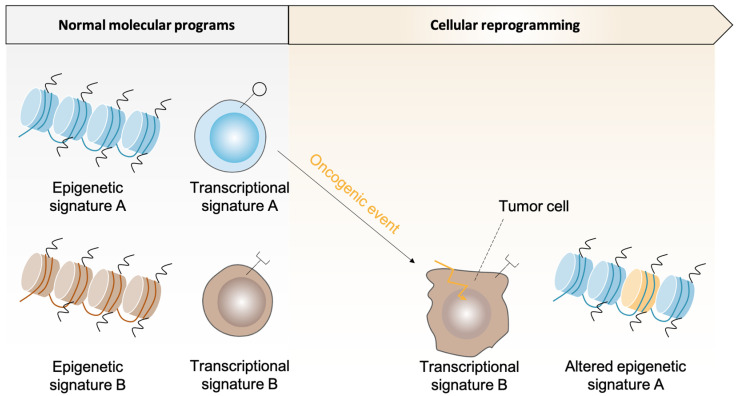
Epigenetic memory as a robust tool to infer cancer cell-of-origin. During tumorigenesis, oncogenic driver events can lead to drastic changes in the transcriptomic landscape of the transformed cell and mechanically modify the histological and morphological properties of the cancer cell compared to its initiating counterpart. Epigenetics comprises robust molecular marks of tissue-specific developmental program that may be retained during cancer initiation and progression.

**Figure 3 ijms-23-06310-f003:**
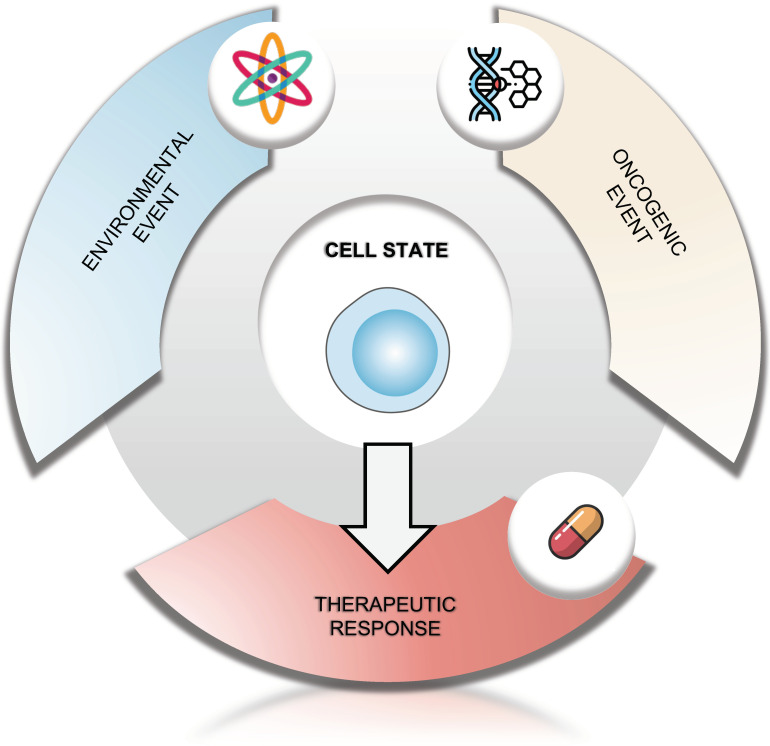
The therapeutic response is conditioned by the combination of the oncogenic event and the environmental factors at the origin of the tumor transformation, and the cellular context in which they occur.

## Data Availability

Not applicable.

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
