# Peer review of "Complex Elucidation of Cells-of-Origin in Pediatric Soft Tissue Sarcoma: From Concepts to Real Life, Hide-and-Seek through Epigenetic and Transcriptional Reprogramming"

_ijms, 2022, doi:10.3390/ijms23116310_

Round 1
Reviewer 1 Report
The authors present here a review on the cell of origin of sarcomas and how it can be masked by epigenetic and transcriptomic reprogramming.
One of the major points is that contrary to what is indicated in the title the authors do not present the current data on "sarcomas" as a whole but only for a few particular and rare types since they are associated with a specific translocation. The authors must then either modify accordingly the title or extend the content of their manuscript.
Then the authors present very little data on sarcomas, but sometimes more of a general review on the mechanisms and ways they could be relevant in sarcomas. For example, point 4 on epigenetics, 80% of which is dedicated to mechanisms and 20% of which deals a little with RMS, is no more than one page long. The authors should then consolidate this part with outstanding work done in sarcomas, different sarcomas, different types of sarcomas. Also paragraph 5 "Clinical relevance...", apart from the work on RMS already mentioned, the paragraph does not directly concern sarcomas and puts forward possible interests in the elucidation of the cell of origin by citing other work in other types of cancer.
Finally, minor points:
-while Figure 1 illustrates the point about RMS, Figure 2 is not helpful.
Additional figures and/or tables could also illustrate/support the text.
-Paragraph 7 is indeed “6”
-l261, the sentence should be rewritten correctly.
-153 references is maybe too much.
Author Response
We thank the reviewer for his/her critical comments that help us to improve the quality of our manuscript.
Taking into account his/her comment, we now have :
1- Modified the title, by integrating the notion that we also wish to develop concepts regarding the definition of soft-tissue sarcomas cells-of-origins.
2- Integrated new examples to illustrate our words, including soft-tissue sarcomas that are not driven by translocation, to illustrate more accurately their heterogeneity (lines 76 ; 116-123 ; 308-316).
3- Consolidated part 4, although to our knowledge, few works have been performed so far to decipher soft-tissue sarcomas' cells-of-origin based on the epigenetic memory concept (lines 308-316).
4- Indicated that we used rhabdomyosarcoma as an example in part 5 (line 76).
We also corrected minor points, except figure 2 that we still find helpful to integrate accurately the concepts developped.
Reviewer 2 Report
The manuscript entitled " Cells-of-origin of soft tissue sarcoma: hide-and-seek through epigenetic and transcriptional reprogramming” provided a systemic review of a large amount of past and recent literatures regarding the concept and identity of cell(s)-of-origin and cellular reprogramming mediated by genetic oncogenic drivers. Role of epigenetic alteration and marks is emphasized and discussed for accurate inference of cell(s)-of-origin to improve the clinical management of patients with soft tissue sarcoma. Overall, it is a nicely written in most parts of manuscript. However, I have following comments/concerns, which need to be addressed before publishing it.
- There are a total of 7 sections. I did not see section 6, which is missed and needs to be added.
- Please cite a new refence for this statement “Soft tissue sarcoma (STS) represent a heterogeneous group of malignant tumors comprising a 36 collection of more than 100 histological subtypes [1]”. Ref 1 has only sated approximately 70 subtypes, which did not support the Authors’ statement.
- There are many references, which may not be cited or missed in the text of manuscript. Please double-check all of references to see whether they are cited properly. E.g. Ref 141, 144-148, 154-162.
- Please check redundant reference papers. For example, Ref 28 and Ref 125 are same, which needs to be corrected.
- Please check thoroughly typos. E.g. in line 308, a period is missed at the end of sentence.
Author Response
We first want to thank the reviewer for his/her supportive comment.
We have improved our manuscript accordingly :
- The number of sections has been corrected ;
- The adequate citation has been added.
- The references have been corrected.
- Redundant references have been eliminated.
- Typos has been corrected.
Round 2
Reviewer 1 Report
I thank authors for considering my comments, nevertheless I still have concerns:
-The review is still focus on pediatric sarcoma (which is not a problem) and it has to be mentioned in the title otherwise it is misleading.
Authors responded to one of my concerns: "Consolidated part 4, although to our knowledge, few works have been performed so far to decipher soft-tissue sarcomas' cells-of-origin based on the epigenetic memory concept (lines 308-316)". Ok but according to the title, it is the subject of the review.
- Figure 2 is not useful, and more figure could illustrate other parts of the papers.
Author Response
We thank again the reviewer for his/her constructive comments.
- Title has been modified.
- The subject of the review is to highlight the complexity and importance of elucidating the cell-of-origin of STS, and this has been more carefully stated by modyfying the title and adding an adequate mention (l76-78).
- A supplementary figure has now been added.
Reviewer 2 Report
I am satisfied with the author's answer. It's a good read now.
Author Response
We thank the reviewer for his/her supportive comments.